materials science/organometallic chemistry/synthetic chemistry

chain shuttling polymerization, nonbridged half-titanocene catalyst, bis(phenoxy-imine) Zr catalyst, block copolymers

**Author for correspondence:**
Dongbing Liu
e-mail: liudb.bjhy@sinopec.com

This article has been edited by the Royal Society of Chemistry, including the commissioning, peer review process and editorial aspects up to the point of acceptance.

# Studies on chain shuttling polymerization reaction of nonbridged half-titanocene and bis(phenoxy-imine) Zr binary catalyst system

Qinwen Xu[1,2], Rong Gao[1,2] and Dongbing Liu[1,3]

[1]Polyolefins National Engineering and Research Center, [2]Polyethylene Research Center, and [3]Institute of Catalysis Science, Sinopec Beijing Research Institute of Chemical Industry, Beijing 100013, People's Republic of China

DL, 0000-0002-5514-0153

In this contribution, olefin block copolymers were produced via chain shuttling polymerization (CSP), using a new combination of catalysts and a chain shuttling agent (CSA) diethylzinc ($ZnEt_2$). The binary catalyst system included nonbridged half-titanocene catalyst, $Cp^*TiCl_2(O\text{-}2,6\text{-}^iPr_2C_6H_3)$ (Cat **A**) and bis(phenoxy-imine) zirconium, $\{\eta^2\text{-}1\text{-}[C(H)=NC_6H_{11}]\text{-}2\text{-}O\text{-}3\text{-}^tBu\text{-}C_6H_3\}_2ZrCl_2$ (Cat **B**), as well as co-catalyst methylaluminoxane (MAO). In contrast to dual-catalyst system in the absence of CSA, the blocky structure was obtained in the presence of CSA and rationalized from rheological studies. The binary catalyst system could cause the CSP reaction to occur in the presence of CSA $ZnEt_2$, which yielded broad distribution ethylene/1-octene copolymers ($M_w/M_n$: 35.86) containing block polymer chains with high $M_w$. The presented dual-catalytic system was applied for the first time in CSP and has a potential to be extended to produce a library of olefin block copolymers that can be used as advanced additives for thermoplastics.

## 1. Introduction

Chain shuttling polymerization (CSP) process involves the shuttling of growing polymer chains via a chain shuttling agent (CSA) between two catalyst active sites, which efficiently prepare block polymers. CSP is based on reversible chain transfer reactions, including catalyzed chain growth [1,2] and coordinative chain transfer polymerization [3–5]. According to the literatures reported by Valente *et al.* [6] and Zinck [7], CSP catalysts can be selected from FI catalyst, pyridylamide catalyst, α-diimine catalyst, rare earth catalyst and metallocene catalyst.

**Scheme 1.** Structures of the nonbridged half-titanocene complex (Cat **A**) and bis(phenoxy-imine) zirconium complex (Cat **B**).

The above-mentioned selection of catalysts could be explored in catalytic production of ethylene-based block copolymers. For example, ethylene/1-octene block copolymers were prepared via CSP using bis(phenoxy-imine) Zr and pyridylamide-Hf catalyst system [8,9], whereas polymerization using a binary zirconocene-based catalyst mixture or a metallocene-Hf catalyst afforded ethylene/1-hexene block copolymers [10,11]. Interestingly, catalyst systems that incorporate late transition metals, such as α-diimine structures, exhibit unique 'chain walking' phenomenon [12–15]. The α-diimine-based catalytic systems were applied to prepare topologies of block polyethylene with linear/branched [16,17], or branched/hyperbranched [18].

A catalytic system can define the microstructure of the resulting polymers, enabling access to the macromolecular structures with new properties. For example, CSP of propylene using either the pyridylamide-Hf catalyst or the binary zirconocene-based catalyst system realized copolymer structures with a broad range of molecular weights [19,20]. Meanwhile, CSP of racemic propylene oxide using bimetallic chromium catalyst resulted in hydroxy-telechelic isotactic polypropylene oxide [21]. Interestingly, blocky polymer structures of styrene/isoprene, or isoprene alone were realized via CSP using rare earth metal-based catalyst system [22–24].

The core of the CSP method is to find a catalytic system where catalysts and chain shuttling agents are well matched. However, to date, there is no specific theory to reach this goal, and a large number of experiments are still required to find an efficient catalyst system for CSP.

Therefore, in this work, we first explored a binary catalytic system, composed of nonbridged half-titanocene (Cat **A**) and bis(phenoxy-imine) zirconium (Cat **B**) catalysts in a CSP reaction of ethylene block copolymers. The binary catalyst system was used to perform ethylene homopolymerization and ethylene/1-octene copolymerization in the presence of co-catalyst methylaluminoxane (MAO) and CSA diethylzinc (ZnEt$_2$). The polymers produced by the dual-catalytic system (Cat **A**/Cat **B**/MAO/ZnEt$_2$) have remarkable characteristics, including broad molecular weight distribution and containing high molecular weight block polymer chains. The block copolymers are capable of improving compatibility of the blends, and thus polymer mixtures containing block polymer chains have superior physical and mechanical properties to general blends [25]. Herein this work provides an exciting avenue to access and develop new advanced material technology.

# 2. Experimental procedure

## 2.1. Materials

Methylaluminoxane (MAO, 10 wt% Al in toluene) was purchased from Albemarle Corporation and used directly without further purification. Diethyl zinc (ZnEt$_2$, 1.5 M in toluene) was obtained from Acros Organics and used as received. Toluene was purchased from Hangzhou Chemical Reagents Company of China and purified by MBRAUN SPS-800X solvent purification system before use. 1-Octene and 1-Hexane were supplied by J&K Chemical Ltd and dried over 4 Å molecular sieves. Polymerization-grade ethylene was obtained from the Sinopec Yanshan Petrochemical Company. All other chemicals were commercially available and used as received.

## 2.2. Complex synthesis and characterization

Complex **A** (Cat **A**, scheme 1) was synthesized by the reaction of Cp*TiCl$_3$ with one equivalent of the lithium 2,6-diisopropylphenolate according to the literature [26]. However, complex **B** (Cat **B**,

scheme 1) was synthesized by the reaction of $ZrCl_4(THF)_2$ with two equivalents of lithium salt of N-(3-tert-butylsalicylidene) cyclohexylamine according to the literature [27].

### 2.2.1. Cp*TiCl$_2$(O-2,6-$^i$Pr$_2$C$_6$H$_3$) (Cat A)

$^1$H NMR (300 MHz, CDCl$_3$, $\delta$, ppm): 1.38 (d, 12H, $J = 9.2$ Hz, (CH$_3$)$_2$CH-), 2.00 (s, 15H, C$_5$(CH$_3$)$_5$), 3.51 (m, 2H, (CH$_3$)$_2$CH-), 7.11–7.16 (m, 3H, C$_6$H$_3$).

### 2.2.2. {$\eta^2$-1-[C(H)=NC$_6$H$_{11}$]-2-O-3-$^t$Bu-C$_6$H$_3$}$_2$ZrCl$_2$ (Cat B)

$^1$H NMR (300 MHz, CDCl$_3$, $\delta$, ppm): 1.60 (s, 18H, t-Bu), 1.15–2.10 (m, 20H, CH$_2$), 3.83–4.07 (m, 2H, CH), 6.90–7.58 (m, 6H, Ar-H), 8.23 (s, 2H, CH=N).

## 2.3. Polymerization procedure

The ethylene polymerization was carried out in a jacketed 1 l high-pressure polymerization reactor (Julabo ACE). The reactor was dried for at least 1 h at 120°C under vacuum. Then the reactor was repeatedly evacuated and purged with nitrogen gas three times, followed by a final purge with ethene at 60°C. Then 500 ml of dried toluene was introduced into the reactor, and the stirring speed was controlled at 300 r.p.m. At the same time, the toluene solutions of catalysts, co-catalyst MAO and CSA ZnEt$_2$ (if needed) were injected into the reactor. Ethylene was fed continuously to maintain the required 1.0 MPa pressure during the reaction. After 30 min, the polymerization was terminated, the resulting mixtures were immediately poured into acidified alcohol (5 vol% of hydrochloric acid). The polymers were washed with an excess of alcohol, and dried in vacuum at 50°C for 4 h until constant weight was attained.

## 2.4. Polymer characterization

Gel permeation chromatography (GPC) was performed with a Polymer Laboratories PL-220 GPC instrument using 1,2,4-trichlorobenzene as an eluent and operated at a flow rate of 1.0 ml min$^{-1}$ and temperature of 150°C.

Differential scanning calorimetry was performed on a Perkin-Elmer 8500 thermal analyser under a nitrogen atmosphere. The polymers of 10 mg were heated to 160°C at the rate of 10°C min$^{-1}$ and remained at 160°C for 1 min. Then, the samples were cooled to 0°C at the rate of 10°C min$^{-1}$ and held at 0°C for 3 min. Finally, the polymers were reheated to 160°C at the rate of 10°C min$^{-1}$. Peak melting temperatures of the polymers were determined from the second heating curves.

$^1$H NMR spectra were recorded on a Bruker TopSpin 300 MHz spectrometer using chloroform-d as a solvent at 25°C. $^{13}$C NMR spectra of the polymers were recorded with a Bruker TopSpin 300 MHz spectrometer using o-dichlorobenzene-d$_4$ (o-C$_6$D$_4$Cl$_2$) as a solvent at 100°C.

The linear viscoelastic properties of all samples were measured using the MCR 302 rotational rheometers with 25 mm parallel disc geometry and a gap of 1 mm. The dynamic frequency sweep tests were conducted between 0.1 and 500 rad s$^{-1}$ with a strain amplitude of 1.25% at 200°C.

# 3. Results and discussion

## 3.1. Dual-catalyst system (Cat A + Cat B)/MAO catalytic ethylene homopolymerization

The effect of CSA on the molecular weight and molecular weight distribution of the polymers was analysed to investigate CSP reaction of the dual-catalytic system (Cat A/Cat B/MAO/ZnEt$_2$). The properties of the resulting polymers are summarized in table 1.

The results of ethylene polymerization using either Cat A or Cat B, and the CSA ZnEt$_2$ are listed in table 1 (Runs 1–4). The ethylene homopolymerization using Cat A (Run 1) yielded polyethylene with high molecular weight and polydispersity, which was consistent with the previous report [26].

In the case of the individual catalyst in the presence of ZnEt$_2$, there was a significant decrease in molecular weight and a little decrease in molecular weight distribution compared to the dual-catalyst system (Cat A + Cat B)/MAO (Run 2, Run 4 versus Run 5 in figure 1). Therefore, it indicated that the chain transfer reaction that occurred between the CSA ZnEt$_2$ and single catalyst was faster compared to the propagation and reversible [28].

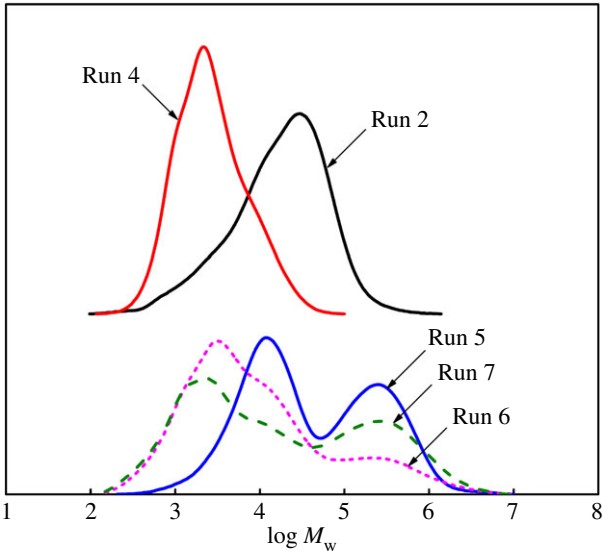

**Figure 1.** The effect of ZnEt$_2$ on the molecular weight and molecular weight distribution of ethylene polymers.

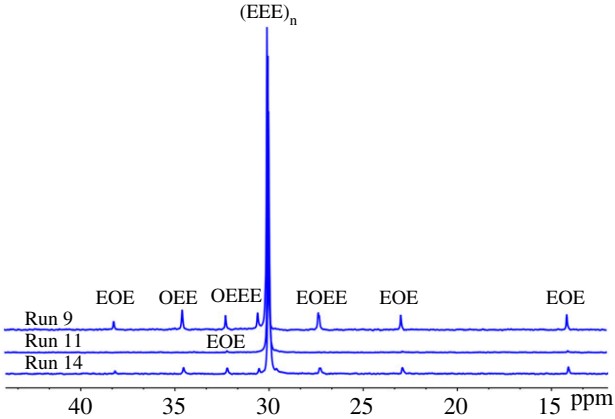

**Figure 2.** $^{13}$C NMR (o-C$_6$D$_4$Cl$_2$, 300 MHz, 100°C) of ethylene/1-octene copolymers.

**Table 1.** The effect of ZnEt$_2$ on ethylene homopolymerization and properties of resulting polyethene.

| Run[a] | Cat **A**/Cat **B** ($\mu$mol/$\mu$mol) | Zn/(Cat **A** + Cat **B**) (molar ratio) | A[b] | $M_w \times 10^{-4}$ (g mol$^{-1}$) | $M_w/M_n$ | $T_m$ (°C) | $\Delta H_f$ (J g$^{-1}$) |
|---|---|---|---|---|---|---|---|
| 1 | 4/0 | 0 | 6.54 | 45.18 | 5.88 | 132.6 | 159.4 |
| 2 | 4/0 | 300 | 4.21 | 3.45 | 5.58 | 132.0 | 225.5 |
| 3 | 0/4 | 0 | 3.47 | 1.20 | 2.91 | 131.9 | 252.4 |
| 4 | 0/4 | 300 | 5.26 | 0.44 | 2.50 | 131.1 | 240.9 |
| 5 | 2/2 | 0 | 4.76 | 17.21 | 15.66 | 134.1 | 217.1 |
| 6 | 2/2 | 300 | 6.32 | 10.28 | 40.16 | 130.6 | 241.2 |
| 7 | 2/2 | 450 | 7.23 | 18.43 | 72.60 | 132.4 | 233.0 |

[a]Conditions: Cat **A** + Cat **B** = 4 $\mu$mol, Al(MAO)/(Cat **A** + Cat **B**) molar ratio = 1500, 500 ml toluene, 10 atm ethylene, $T_p$ = 60°C, $t_p$ = 30 min.
[b]Activity: 10$^6$ g mol$^{-1}$ h$^{-1}$.

When using the binary catalyst system (Cat **A** + Cat **B**)/MAO, the amount of ZnEt$_2$ had a significant effect on molecular weight and molecular weight distribution of the obtained polymers (Runs 5–7 in table 1 and figure 1). Compared with polymers produced by a single catalyst, the polydispersity of

polymers produced by the dual-catalytic system (Cat **A**/Cat **B**/MAO/ZnEt$_2$) was broadened with increasing the amount of ZnEt$_2$. The fractions with lower $M_w$ showed bimodal distributions, while the fragments with higher $M_w$ had unimodal distributions (Runs 6–7 in figure 1). There was no significant chain transfer reaction between either catalyst in the absence of ZnEt$_2$. However, the chain transfer efficiency between ZnEt$_2$ and Cat **A** or Cat **B** was different. This difference would lead to more complicated chain transfer reactions, which would result in poor control of polymers' architecture.

On the one hand, the β-elimination process occurred to generate inherently low molecular weight polymers terminated with vinyl group in the catalysis performed with Cat **B**. Catalyst **A** had good copolymerization activity for α-olefins (including the generated macromonomers). Therefore, it would be possible that macromonomers with vinyl group as comonomer were inserted into growing polymer A. However, this process would be difficult due to the large steric hindrance of the comonomer. On the other hand, the growing polymer chains could be easily displaced with an ethyl group from ZnEt$_2$ rather than terminated by the β-elimination process. However, the content of higher molecular weight components rose with increasing the ZnEt$_2$ amount (Runs 6–7 in figure 1). Hence, it was impossible that the fractions with higher molecular weight were the polymers with macromolecular long-chain branches.

It would be reasonable to conclude that the fractions with lower $M_w$ were the mixtures that were, respectively, produced by the Cat **A**/MAO/ZnEt$_2$ and Cat **B**/MAO/ZnEt$_2$ system. While the fractions with higher $M_w$ originated from CSP [16] carried out with the dual-catalytic system (Cat **A**/Cat **B**/MAO/ZnEt$_2$). The amount of ZnEt$_2$ had a negligible impact on the thermal behaviour of the polymers obtained by dual-catalytic system (see electronic supplementary material, figure S1).

In summary, using the dual-catalyst system (Cat **A** + Cat **B**)/MAO in the presence of ZnEt$_2$ could cause the occurrence of ethylene CSP, which produced polymers that had a broad molecular weight distribution and contained high molecular weight chains.

## 3.2. Dual-catalyst system (Cat **A** + Cat **B**)/MAO catalytic ethylene/1-octene copolymerization

The ethylene/1-octene copolymerization with the dual-catalytic system (Cat **A**/Cat **B**/MAO/ZnEt$_2$) was investigated by introducing 1-octene in order to prepare block copolymers containing branching structure.

### 3.2.1. The effect of ZnEt$_2$ on properties and microstructure of ethylene/1-octene copolymers

CSP was carried out with the dual-catalytic system (Cat **A**/Cat **B**/MAO/ZnEt$_2$) and the architecture (linear/branched) of resulting copolymers was investigated. The results, summarized in table 2, show that the amount of ZnEt$_2$ affected molecular weight, polydispersity, melting point, melting enthalpy and rheological properties of the copolymers.

It is possible to judge whether we get block copolymers by comparing their melting point, degree of branching [29] and rheological properties [30,31].

The ternary sequence structure distribution of the ethylene/1-octene copolymers are listed in table 3. The E represents an ethylene unit, while O represents an octene unit. The EEO appeared only in these spectra (Runs 9 and 14 in figure 2), indicating that this sequence was only present in copolymers with high contents of 1-octene. The EEO ternary sequence content produced by the dual-catalytic system (Cat **A**/Cat **B**/MAO/ZnEt$_2$) was decreased to 0.042 (Run 14) compared with EEO content got from Cat **A** (0.102, Run 9). Herein it indicated that the dual-catalyst system (Cat **A**+ Cat **B**)/MAO yielded new polymer chains in the presence of ZnEt$_2$. The detailed NMR analysis of samples is shown in the electronic supplementary material, figures S2, S3 and S4.

Based on figure 3, the molecular weight of polymers obtained from Runs 12 and 14 was near (in the absence and presence of CSA, respectively). Due to the copolymerization ability of Cat **A**, the vinyl group-terminated macromonomers produced by Cat **B** was possible to be inserted into growing polymer A chains, which generated the polymers with macromolecular long-chain branches. Hence, it would be reasonable to conclude that the fractions with high $M_w$ were either polymers with macromolecular long-chain branches or block copolymers originated from CSP. The viscosity of polymers obtained in Run 14 was higher than Run 12 in the low angular frequency range. In the meantime, the shear thinning behaviour of polymers obtained in Run 14 was more pronounced than Run 12 with increasing angular frequency. Hence, it indicated that it was impossible for a large number of polymers with macromolecular long-chain branches to exist in the polymers obtained in Run 14. However, the viscosity of polymers obtained in Run 14 was lower than Run 12

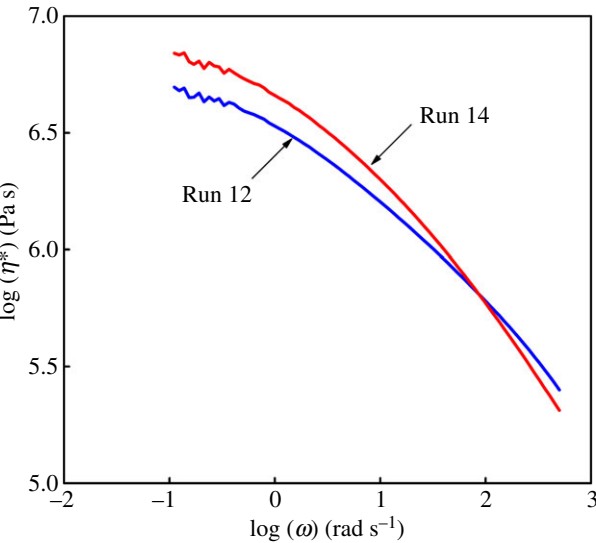

**Figure 3.** The complex viscosity curve of polymers (Runs 12 and 14) at 200°C.

**Table 2.** The effect of ZnEt$_2$ on ethylene/1-octene copolymerization and properties of resulting copolymers.

| Run[a] | Cat **A**/Cat **B** (μmol/μmol) | Zn/(Cat **A** + Cat **B**) (molar ratio) | A[b] | $M_w \times 10^{-4}$ (g mol$^{-1}$) | $M_w/M_n$ | $T_m$ (°C) | $\Delta H_f$ (J g$^{-1}$) | Content of 1-octene[c] |
|---|---|---|---|---|---|---|---|---|
| 8 | 4/0 | 0 | 5.71 | 6.32 | 2.60 | 85.12 | 55.10 | — |
| 9 | 4/0 | 300 | 5.19 | 2.89 | 2.42 | 94.80 | 84.91 | 5.08 |
| 10 | 0/4 | 0 | 4.58 | 1.25 | 2.14 | 131.2 | 244.5 | — |
| 11 | 0/4 | 300 | 7.71 | 1.17 | 5.32 | 130.6 | 240.0 | 0.26 |
| 12 | 2/2 | 0 | 7.45 | 8.74 | 7.71 | 127.7 | 104.0 | — |
| 13 | 2/2 | 150 | 4.95 | 12.45 | 35.13 | 126.7 | 157.9 | — |
| 14 | 2/2 | 300 | 6.89 | 11.93 | 35.86 | 125.8 | 159.1 | 2.12 |
| 15 | 2/2 | 450 | 6.53 | 1.12 | 7.26 | 127.0 | 220.7 | — |

[a]Conditions: 20 ml 1-octene, Cat **A** + Cat **B** = 4 μmol, Al(MAO)/(Cat **A** + Cat **B**) molar ratio = 1500, 500 ml toluene, 10 atm ethylene, $T_p$ = 60°C, $t_p$ = 30 min.
[b]Activity: 10$^6$ g mol$^{-1}$ h$^{-1}$.
[c]In mol% determined by $^{13}$C NMR spectra.

**Table 3.** The triad sequence distributions of ethylene/1-octene copolymers obtained by $^{13}$C NMR analysis.

| Run | EEE | EEO | OEO | EOE | EOO | OOO |
|---|---|---|---|---|---|---|
| 9 | 0.847 | 0.102 | 0 | 0.051 | 0 | 0 |
| 11 | 0.993 | 0 | 0 | 0.007 | 0 | 0 |
| 14 | 0.937 | 0.042 | 0 | 0.021 | 0 | 0 |

in the high angular frequency range. It indicated that the number of hexyl-branches of polymers obtained from Run 14 was lower than Run 12. Thus, I further believed that the polymers obtained from Run 14 existed as block copolymers [32] rather than macromolecular long-chain branched polymers.

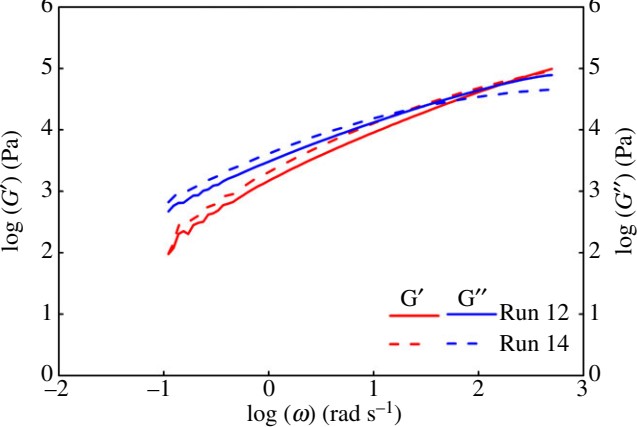

**Figure 4.** The dynamic modulus curve of polymers (Runs 12 and 14) at 200°C.

Next, we investigated dynamic modulus as a parameter of block copolymers. Based on the block copolymers increasing interfacial elasticity of the blends, the dynamic modulus of mixtures increased in the low angular frequency range, whereas the dynamic modulus was near in the high angular frequency range [32]. The storage modulus of polymers obtained in the presence of CSA (Run 14) was higher than in its absence (Run 12). As the angular frequency increased, the storage modulus of polymers obtained in Run 14 increased more slowly than Run 12. It indicated that the block copolymer microstructure existed in polymers obtained with a dual-catalyst system in the presence of $ZnEt_2$ (Run 14 in figure 4).

The effectiveness of copolymerization reaction produces copolymers with significantly decreased molecular weight compared with homopolymers [33]. The addition of 1-octene had a significantly different influence on properties of the polymers obtained by either Cat **A** or Cat **B**, respectively (see electronic supplementary material, figures S5 and S6). After adding 20 ml 1-octene (Run 8), the molecular weight of copolymers obtained with Cat **A** was significantly decreased ($M_w$: 451 800 g mol$^{-1}$ $\rightarrow$ 63 200 g mol$^{-1}$), and the molecular weight distribution became narrowed ($M_w/M_n$: 5.88 $\rightarrow$ 2.60) compared to ethylene homopolymers (Run 1). When catalyst **A** was used, the melting point and melting enthalpy of copolymers were significantly reduced compared to ethylene homopolymers. (Run 1 versus Run 8; $T_m$: 132.6°C $\rightarrow$ 85.12°C; $\Delta H_f$: 159.4 J g$^{-1}$ $\rightarrow$ 55.10 J g$^{-1}$), which indicated that the catalyst **A** had a high copolymerization activity [34,35]. However, in the case of Cat **B**, adding 20 ml 1-octene had little effect on molecular weight, polydispersity, melting point and melting enthalpy of the polymers (Run 3 versus Run 10), indicating that the catalyst **B** was poor effective for copolymerization [36].

According to Runs 8–11 in table 2, adding $ZnEt_2$ reduced molecular weight and polydispersity of the polymers obtained with Cat **A**, in contrast to increased molecular weight distribution in the case of Cat **B** ($M_w/M_n$: 2.14 $\rightarrow$ 5.32). The addition of $ZnEt_2$ also had a different influence on the thermal behaviour of the polymers obtained by either Cat **A** or Cat **B** respectively (see electronic supplementary material, figure S7). Based on GPC data (figure 5), the polymers obtained with Cat **B** (Run 11) were bimodal and incorporated low molecular weight peak. It would be reasonable to conclude that the low molecular weight peak was polymers produced by Cat **B**/MAO/$ZnEt_2$ system.

The chain transfer reaction reduced the molecular weight of the polymers obtained with a single catalyst system. Therefore, in general, the addition of CSA $ZnEt_2$ causes molecular weight and polydispersity of polymers to decrease. When ethylene/1-octene copolymerization was carried out in the presence of the dual-catalyst system (Cat **A** + Cat **B**)/MAO, the effect of amount of $ZnEt_2$ on molecular weight and polydispersity of the polymers was significantly different from previous reports [8]. As the amount of $ZnEt_2$ increased, the molecular weight distribution of polymers gradually widened (Runs 12–14 in table 2). At the same time, the content of components with higher molecular weight increased (Runs 13–14 in figure 6). We thought it would be reasonable to conclude that the fractions with lower $M_w$ were the mixtures produced by individual catalytic systems (Cat **A**/MAO/$ZnEt_2$) and (Cat **B**/MAO/$ZnEt_2$). While the fractions with higher $M_w$ were produced by dual-catalytic system (Cat **A**/Cat **B**/MAO/$ZnEt_2$) via CSP (Runs 13–14 in figure 6) [16]. Further increasing the amount of CSA $ZnEt_2$ (Run 15), lowered the molecular weight of the polymers. This result is consistent with previous reports [37,38], where an increased amount of $ZnEt_2$ in the presence of

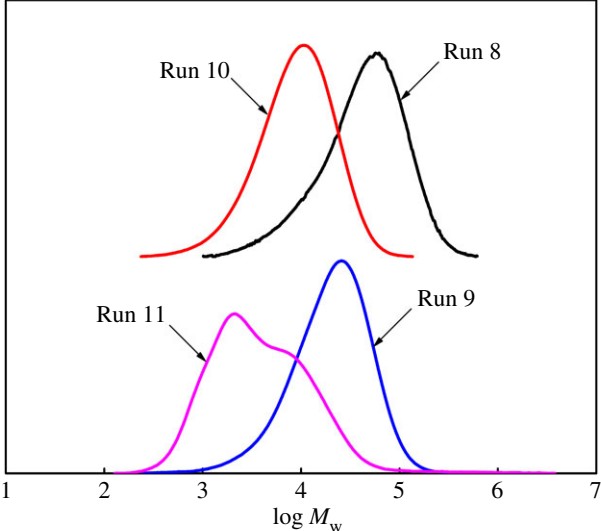

**Figure 5.** The effect of ZnEt$_2$ on the molecular weight and polydispersity of copolymers obtained with individual catalysts.

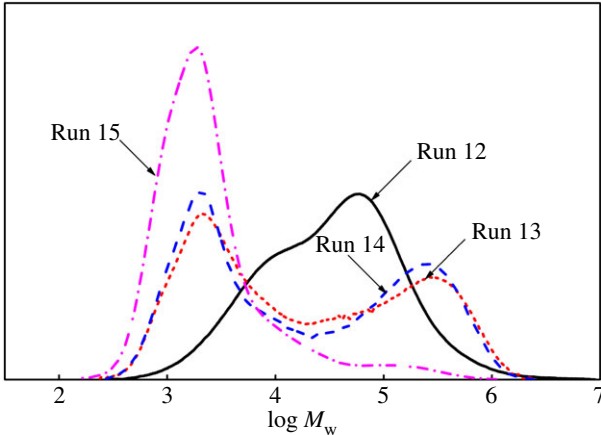

**Figure 6.** The effect of the amount of ZnEt$_2$ on the molecular weight and polydispersity of copolymers obtained with the dual-catalyst system.

dual-catalyst system, led to the increased number of produced polymer chains. It would lead to significant decrease in molecular weight of the polymers.

Using the binary catalyst system (Cat **A** + Cat **B**)/MAO in ethylene homopolymerization produced polymers with large molecular weight and broad molecular weight distribution ($M_w$: 184 300 g mol$^{-1}$, $M_w/M_n$: 72.60) (Run 7) at the Zn/(Cat **A** + Cat **B**) molar ratio of 450. However, ethylene/1-octene copolymerization under the same catalytic conditions led to lower molecular weight and relatively narrower molecular weight distribution of resulting polymers ($M_w$: 11 200 g mol$^{-1}$, $M_w/M_n$: 7.26) (Run 15). These findings were reinforced by previous reports [10,33], where copolymerization not only significantly reduced the molecular weight of the polymers produced by a single catalyst [33], but also decreased the molecular weight of the block copolymers produced by dual-catalytic system [10].

For random copolymers, the melting point is inversely proportional to the comonomer content. As the amount of ZnEt$_2$ increased, the melting point of polymers produced by the binary catalyst system dropped. However, it was still higher than 125°C (Runs 12–14 in table 2 and figure 7). Hence, the crystallization ability of the block copolymers contained in polymers was between the copolymer A and the copolymer B. While their melting enthalpy increased with increasing the amount of CSA. Thus, we concluded that the content and crystallization degree of the block copolymers was high in products of a binary catalyst polymerization. When the Zn/(Cat **A** + Cat **B**) molar ratio was increased to 450 (Run 15), a new melting peak appeared in the polymer melting curve at 115°C. This peak indicated that increasing amount of CSA led to the enhancement of soft segment content in the block copolymers.

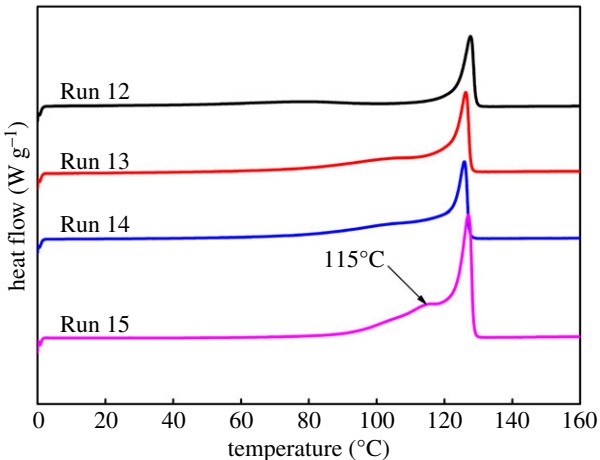

**Figure 7.** The effect of the amount of $ZnEt_2$ on the melting point and melting enthalpy of copolymers produced with the binary catalyst system.

**Table 4.** The effect of catalyst molar ratio Cat **A**/Cat **B** on ethylene/1-octene copolymerization and properties of resulting copolymers.

| Run[a] | Cat **A**/Cat **B** ($\mu$mol/$\mu$mol) | Zn/(Cat **A** + Cat **B**) (molar ratio) | A[b] | $M_w \times 10^{-4}$ (g mol$^{-1}$) | $M_w/M_n$ | $T_m$ (°C) | $\Delta H_f$ (J g$^{-1}$) |
|---|---|---|---|---|---|---|---|
| 16 | 1/3 | 300 | 6.97 | 0.81 | 3.16 | 128.6 | 247.0 |
| 14 | 2/2 | 300 | 6.89 | 11.93 | 35.86 | 125.8 | 159.1 |
| 17 | 3/1 | 300 | 2.39 | 1.30 | 6.58 | 124.4 | 179.2 |

[a]Conditions: 20 ml 1-octene, Cat **A** + Cat **B** = 4 $\mu$mol, Al(MAO)/(Cat **A** + Cat **B**) molar ratio = 1500, 500 ml toluene, 10 atm ethylene, $T_p$ = 60°C, $t_p$ = 30 min.
[b]Activity: $10^6$ g mol$^{-1}$ h$^{-1}$.

In summary, the dual-catalyst system (Cat **A** + Cat **B**)/MAO was used in ethylene/1-octene copolymerization in the presence of CSA $ZnEt_2$, which produced the polymers with weight-average molecular weight up to 119 300 g mol$^{-1}$ and molecular weight distribution up to 35.86. At the same time, the high molecular weight block polymer chains were contained in the ethylene/1-octene copolymers.

### 3.2.2. The effect of Cat **A**/Cat **B** molar ratio on the copolymer properties and structure

The effect of Cat **A**/Cat **B** molar ratio on the properties of the polymers prepared by the dual-catalytic system (Cat **A**/Cat **B**/MAO/$ZnEt_2$) was investigated by comparing molecular weight, polydispersity, melting point and melting enthalpy of the resulting polymers. The findings are summarized in table 4.

As the molar ratio of the Cat **A**/Cat **B** in the binary catalyst system increased, the molecular weight of polymer mixtures rose and then decreased. The molecular weight of block copolymers followed the same trend (table 4 and figure 8). When the molar ratio of Cat **A**/Cat **B** was 2/2 (Run 14), the molecular weight of block copolymers was high. This outcome was rationalized by the improvement of the soft segment content of block copolymers [11]. The catalyst molar ratio reached 3/1 (Run 17), where more significant chain transfer reaction between catalyst **A** and $ZnEt_2$ perhaps occurred. Therefore, reducing the average block length in the resulting polymers [38], which led to a reduction of molecular weight of the block copolymers produced with the dual-catalytic system.

When the molar ratio of the Cat **A**/Cat **B** in the dual-catalytic system increased, the melting point and melting enthalpy of polymers decreased (table 4 and figure 9). When the catalyst molar ratio reached Cat **A**/Cat **B** = 3/1 (Run 17), the polymer melting curve emerged with a new melting peak at 108°C. This occurrence indicated that the soft segment content in the block copolymers is high at the high molar ratio of Cat **A**/Cat **B**.

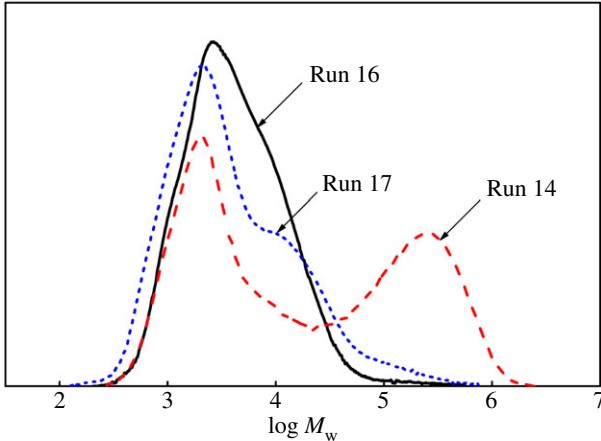

**Figure 8.** The Effect of catalyst molar ratio Cat **A**/Cat **B** in the binary catalytic system on the molecular weight and molecular weight distribution of copolymers.

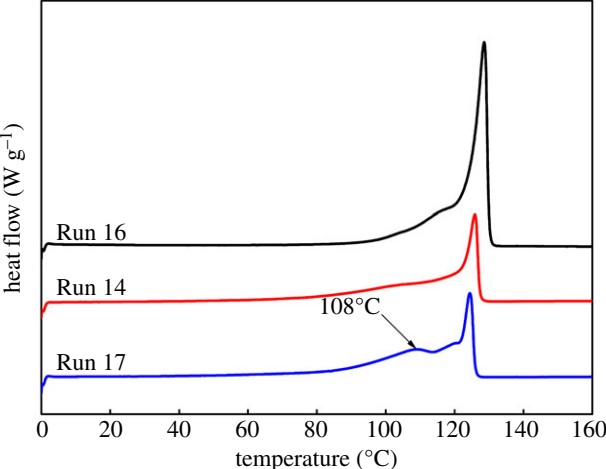

**Figure 9.** The effect of catalyst ratio Cat **A**/Cat **B** in the binary catalytic system on the melting point and melting enthalpy of copolymers.

In summary, the molecular weight, melting point and melting enthalpy of the polymers prepared by the binary catalytic system could be adjusted by changing the catalyst molar ratio. The molecular weight and thermal properties of block copolymers contained in the polymer mixtures also could be adjusted by varying the catalyst molar ratio.

## 4. Conclusion

The dual-catalyst system composed of nonbridged half-titanocene catalyst (Cat **A**) and bis(phenoxy-imine) zirconium catalyst (Cat **B**) was used in ethylene/1-octene copolymerization in the presence of co-catalyst MAO and CSA $ZnEt_2$. The obtained polymers had a weight-average molecular weight up to 119 300 g mol$^{-1}$ and molecular weight distribution up to 35.86, containing block polymer chains with high $M_w$.

The use of the dual-catalytic system (Cat **A**/Cat **B**/MAO/$ZnEt_2$) will be further expanded to CSP of ethylene and other α-olefins, to prepare a series of broad distribution polymers containing blocky structure. These polymers are planned to be used as additives to thermoplastics, improving the physical and mechanical properties of the blends. Herein this work provides a new platform for the development of advanced functional materials.

Data accessibility. Our datasets supporting this article have been uploaded as part of the electronic supplementary material.

Authors' contributions. Q.X. performed the overall experimental work and drafted the manuscript. R.G. participated in the design of the study and coordinated the study. D.L. conceived the study, design of the study and revised manuscript. All authors gave final approval publication.

Competing interests. We declare that we have no competing interests.

Funding. This research received no external funding.

Acknowledgements. We thank Sinopec for permission to publish this original research article.

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
