## [Reviewer comments · Royal Society Open Science]

Review History

RSOS-182007.R0 (Original submission)

Review form: Reviewer 1

Is the manuscript scientifically sound in its present form?

Yes

Are the interpretations and conclusions justified by the results?

Yes

Is the language acceptable?

Yes

Is it clear how to access all supporting data?

Yes

Do you have any ethical concerns with this paper?

No

Have you any concerns about statistical analyses in this paper?

No

Recommendation?

Accept with minor revision (please list in comments)

Comments to the Author(s)

Very interesting work, dealing with the chain shuttling polymerization reaction for the copolymerization of ethylene/1-octene in the presence of the various catalyst. The work can be suggested for publication since no serious scientific issues were found and also the method advances the synthesis of such type of copolymers.

A few remarks:

- I am a little bit astonished why the authors proof the synthesis of block copolymers by rheology and not by NMR. Was it not possible due to the solubility of the synthesized copolymers? Please state such a comment. A hind is that maybe it could be possible to perform HR-MAS experiments if the copolymers can be swollen in a deuterated solvent.

- Was it possible to regain catalysts after the synthesis? This would be extremely interesting since reusing of the catalyst for similar synthesis can save a lot on resources.

- I can understand that higher molecular weight is kind of important but since the dispersity index is high, can it be considered that the properties (e.g., the mechanical properties) of the synthesized polymers are homogeneous?

- Was it possible to perform somehow mechanical tests on the synthesized samples? A DMTA experiment would be interesting in order to approximately estimate the young modulus of the copolymers.

- Please check for language and typo error. In some parts, the language is not nice and there are also mistakes and mix up such us "block polymers" or expressions like "In order to investigate whether the catalytic system (A/B/MAO/ZnEt₂) could prepare block copolymers with branching structure by chain shuttling polymerization. It was investigated that the amount of ZnEt₂ affected on molecular weight, molecular weight distribution, ..."

Review form: Reviewer 2

Is the manuscript scientifically sound in its present form?

Yes

Are the interpretations and conclusions justified by the results?

Yes

Is the language acceptable?

Yes

Is it clear how to access all supporting data?

Not Applicable

Do you have any ethical concerns with this paper?

Yes

Have you any concerns about statistical analyses in this paper?

No

Recommendation?

Accept with minor revision (please list in comments)

Comments to the Author(s)

This research studies the chain shuttling polymerization reaction of bisphenoxy-imine zr binary catalyst systems and used a few characterization methods to study the material structures and properties.

1. the introduction was not comprehensively reviewing the state of the art;
2. for the DSC tests, did the test get rid of the thermal history of the materials?
3. for figure 3 & 4, what do the cross points of $\log G'$ and $\log G''$ mean?

Decision letter (RSOS-182007.R0)

14-Jan-2019

Dear Dr Liu:

Title: Studies on chain shuttling polymerization reaction of nonbridged half-titanocene and bis(phenoxy-imine) Zr binary catalyst system
Manuscript ID: RSOS-182007

Thank you for submitting the above manuscript to Royal Society Open Science. On behalf of the Editors and the Royal Society of Chemistry, I am pleased to inform you that your manuscript will be accepted for publication in Royal Society Open Science subject to minor revision in accordance with the referee suggestions. Please find the reviewers' comments at the end of this email.

The reviewers and handling editors have recommended publication, but also suggest some minor revisions to your manuscript. Therefore, I invite you to respond to the comments and revise your manuscript.

Please also include the following statements alongside the other end statements. As we cannot publish your manuscript without these end statements included, if you feel that a given heading is not relevant to your paper, please nevertheless include the heading and explicitly state that it is not relevant to your work. We have included a screenshot example of the end statements for reference.

- Ethics statement

Please clarify whether you received ethical approval from a local ethics committee to carry out your study. If so please include details of this, including the name of the committee that gave consent in a Research Ethics section after your main text. Please also clarify whether you received informed consent for the participants to participate in the study and state this in your Research Ethics section.

OR

Please clarify whether you obtained the necessary licences and approvals from your institutional animal ethics committee before conducting your research. Please provide details of these licences and approvals in an Animal Ethics section after your main text.

OR

Please clarify whether you obtained the appropriate permissions and licences to conduct the fieldwork detailed in your study. Please provide details of these in your methods section.

Because the schedule for publication is very tight, it is a condition of publication that you submit the revised version of your manuscript before 23-Jan-2019. Please note that the revision deadline will expire at 00.00am on this date. If you do not think you will be able to meet this date please let me know immediately.

Best wishes,
Dr Laura Smith
Publishing Editor, Journals

On behalf of the Subject Editor Professor Anthony Stace and the Associate Editor Professor Eva Hevia.

RSC Associate Editor:
Comments to the Author:
(There are no comments.)

RSC Subject Editor:
Comments to the Author:
(There are no comments.)

Reviewer comments to Author:
Reviewer: 1

Comments to the Author(s)

Very interesting work, dealing with the chain shuttling polymerization reaction for the copolymerization of ethylene/1-octene in the presence of the various catalyst. The work can be suggested for publication since no serious scientific issues were found and also the method advances the synthesis of such type of copolymers.

A few remarks:

- I am a little bit astonished why the authors proof the synthesis of block copolymers by rheology and not by NMR. Was it not possible due to the solubility of the synthesized copolymers? Please state such a comment. A hind is that maybe it could be possible to perform HR-MAS experiments if the copolymers can be swollen in a deuterated solvent.
- Was it possible to regain catalysts after the synthesis? This would be extremely interesting since reusing of the catalyst for similar synthesis can save a lot on resources.
- I can understand that higher molecular weight is kind of important but since the dispersity index is high, can it be considered that the properties (e.g., the mechanical properties) of the synthesized polymers are homogeneous?
- Was it possible to perform somehow mechanical tests on the synthesized samples? A DMTA experiment would be interesting in order to approximately estimate the young modulus of the copolymers.
- Please check for language and typo error. In some parts, the language is not nice and there are also mistakes and mix up such as "block polymers" or expressions like "In order to investigate whether the catalytic system (A/B/MAO/ZnEt₂) could prepare block copolymers with branching structure by chain shuttling polymerization. It was investigated that the amount of ZnEt₂ affected on molecular weight, molecular weight distribution, ..."

Reviewer: 2

Comments to the Author(s)

This research studies the chain shuttling polymerization reaction of bisphenoxy-imine zirconium binary catalyst systems and used a few characterization methods to study the material structures and properties.

1. the introduction was not comprehensively reviewing the state of the art;
2. for the DSC tests, did the test get rid of the thermal history of the materials?
3. for figure 3 & 4, what do the cross points of $\log G'$ and $\log G''$ mean?

Author's Response to Decision Letter for (RSOS-182007.R0)

See Appendix A.

RSOS-182007.R1 (Revision)

Review form: Reviewer 1

Is the manuscript scientifically sound in its present form?

Yes

Are the interpretations and conclusions justified by the results?

Yes

Is the language acceptable?

Yes

Is it clear how to access all supporting data?

Yes

Do you have any ethical concerns with this paper?

No

Have you any concerns about statistical analyses in this paper?

No

Recommendation?

Accept as is

Comments to the Author(s)

The revised manuscript can be accepted for publication.

Review form: Reviewer 2

Is the manuscript scientifically sound in its present form?

Yes

Are the interpretations and conclusions justified by the results?

Yes

Is the language acceptable?

Yes

Is it clear how to access all supporting data?

Not Applicable

Do you have any ethical concerns with this paper?

No

Have you any concerns about statistical analyses in this paper?

No

Recommendation?

Accept as is

Comments to the Author(s)

Accept

Decision letter (RSOS-182007.R1)

08-Feb-2019

Dear Dr Liu:

Title: Studies on chain shuttling polymerization reaction of nonbridged half-titanocene and bis(phenoxy-imine) Zr binary catalyst system

Manuscript ID: RSOS-182007.R1

It is a pleasure to accept your manuscript in its current form for publication in Royal Society Open Science. The chemistry content of Royal Society Open Science is published in collaboration with the Royal Society of Chemistry.

Yours sincerely,

Dr Laura Smith

Publishing Editor, Journals

On behalf of the Subject Editor Professor Anthony Stace and the Associate Editor Professor Eva Hevia.

RSC Associate Editor:
Comments to the Author:
(There are no comments.)

RSC Subject Editor:
Comments to the Author:
(There are no comments.)

Reviewer(s)' Comments to Author:
Reviewer: 2

Comments to the Author(s)
accept

Reviewer: 1

Comments to the Author(s)
The revised manuscript can be accepted for publication.

Appendix A

Dear Editors and Reviewers:

Thank you for your letter and for the reviewers' comments concerning our manuscript entitled "Studies on chain shuttling polymerization reaction of nonbridged half-titanocene and bis(phenoxy-imine) Zr binary catalyst system" (ID: RSOS-182007). Those comments are all valuable and very helpful for revising and improving our paper, as well as the important guiding significance to our researches. We have studied comments carefully and have made correction which we hope meet with approval. Revised portion are marked in purple in the paper. The main corrections in the paper and the responds to the reviewer's comments are as flowing:

Responds to the reviewer's comments:

Reviewer: 1

1. Why the authors proof the synthesis of block copolymers by rheology and not by NMR?

Response: On one hand, the carbon nuclear magnetic characterization only could obtain the average number of branches and the distribution of branches on the polymer chain. This characterization results couldn't directly prove existence of block copolymers. On the other hand, the block structure could significantly affect the mechanical properties of the blends. Therefore, we chose rheology rather than NMR to prove the synthesis of block copolymers.

2. Was it possible to perform HR-MAS experiment?

Response: Because the copolymers dissolved in deuterated trichlorobenzene rather than swelling, it couldn't perform HR-MAS experiments.

3. Was it possible to regain catalysts after the synthesis?

Response: Because the olefin chain shuttling polymerization experiments was performed in solution, it was possible to regain catalysts in theory. Since the catalyst was easily inactivated, it would lead to expensive regaining cost of the catalysts. Therefore, we generally didn't choose to recycle this type of catalysts.

4. Can it be considered that the properties (e.g., the mechanical properties) of the synthesized polymers are homogeneous?

Response: Since the catalytic system produced polymer mixtures containing block copolymers, whereas the block copolymers had a compatibilization effect on the mixtures. Therefore, the properties of the polymers with high polydispersity were still homogeneous.

5. Was it possible to perform somehow mechanical tests on the synthesized samples, such as DMTA experiment?

Response: We believed that the DMTA test could provide valuable young's modulus information. However, the amount of some experimental samples was quite low. It was difficult to make standard testing samples according to DMTA test requirements. Therefore, we only regretted to say that DMTA characterization couldn't be performed for some significant experimental samples.

making it impossible to prepare test strips in accordance with DMTA test requirements.

6. Please check for language and typo error.

Response: The detailed modify of language and typo error see purple text in new version manuscript.

Reviewer: 2

1. The introduction was not comprehensively reviewing the state of the art.

Response: In the course of this research, we have reviewed a quite literatures. Although only the most related literature was added into introduction, the introduction has fully covered all the literatures, especially for the latest reports, on chain shuttling polymerization research.

2. For the DSC tests, did the test get rid of the thermal history of the materials?

Response: When performing the DSC test, the sample was heated to 160 °C at a rate of 10°C min⁻¹ and keep it for 1 minute. The operation process could completely eliminate the thermal history of the samples.

3. For figure 3 &4, what do the cross points of logG' and logG'' mean?

Response: The cross point of log G' and log G'' is the flow point. In the low angular frequency range, the storage modulus of the polymers is lower than the loss modulus, indicating that the polymers viscosity is dominant on the range. As the angular frequency increases, the storage modulus of the polymers increases rapidly, indicating the polymers elasticity increasing.

We tried our best to improve the manuscript and made large language changes in the manuscript. These changes will not influence the content and framework of the paper. And here we did not list the changes but marked in purple in revised paper.

We appreciate for Editors/Reviewers' warm work earnestly, and hope that the correction will meet with approval.

Once again, thank you very much for your comments and suggestions.